# The Scaling Limit of High-Dimensional Online Independent Component Analysis

**Chuang Wang and Yue M. Lu**
John A. Paulson School of Engineering and Applied Sciences
Harvard University
33 Oxford Street, Cambridge, MA 02138, USA
{chuangwang,yuelu}@seas.harvard.edu

## Abstract

We analyze the dynamics of an online algorithm for independent component analysis in the high-dimensional scaling limit. As the ambient dimension tends to infinity, and with proper time scaling, we show that the time-varying joint empirical measure of the target feature vector and the estimates provided by the algorithm will converge weakly to a deterministic measured-valued process that can be characterized as the unique solution of a nonlinear PDE. Numerical solutions of this PDE, which involves two spatial variables and one time variable, can be efficiently obtained. These solutions provide detailed information about the performance of the ICA algorithm, as many practical performance metrics are functionals of the joint empirical measures. Numerical simulations show that our asymptotic analysis is accurate even for moderate dimensions. In addition to providing a tool for understanding the performance of the algorithm, our PDE analysis also provides useful insight. In particular, in the high-dimensional limit, the original coupled dynamics associated with the algorithm will be asymptotically "decoupled", with each coordinate independently solving a 1-D effective minimization problem via stochastic gradient descent. Exploiting this insight to design new algorithms for achieving optimal trade-offs between computational and statistical efficiency may prove an interesting line of future research.

## 1 Introduction

Online learning methods based on stochastic gradient descent are widely used in many learning and signal processing problems. Examples includes the classical least mean squares (LMS) algorithm [1] in adaptive filtering, principal component analysis [2, 3], independent component analysis (ICA) [4], and the training of shallow or deep artificial neural networks [5–7]. Analyzing the convergence rate of stochastic gradient descent has already been the subject of a vast literature (see, *e.g.*, [8–11].) Unlike existing work that analyze the behaviors of the algorithms in *finite dimensions*, we present in this paper a framework for studying the exact dynamics of stochastic gradient algorithms in the *high-dimensional limit*, using online ICA as a concrete example. Instead of minimizing a generic function as considered in the optimization literature, the stochastic algorithm we analyze here is solving an estimation problem. The extra assumptions on the ground truth (*e.g.*, the feature vector) and the generative models for the observations allow us to obtain the exact asymptotic dynamics of the algorithms.

As the main result of this work, we show that, as the ambient dimension $n \to \infty$ and with proper time-scaling, the time-varying joint empirical measure of the true underlying independent component $\boldsymbol{\xi}$ and its estimate $\boldsymbol{x}$ converges weakly to the unique solution of a nonlinear partial differential equation (PDE) [see (6).] Since many performance metrics, such as the correlation between $\boldsymbol{\xi}$ and

$\boldsymbol{x}$ and the support recover rate, are functionals of the joint empirical measure, knowledge about the asymptotics of the latter allows us to easily compute the asymptotic limits of various performance metrics of the algorithm.

This work is an extension of a recent analysis on the dynamics of online sparse PCA [12] to more general settings. The idea of studying the scaling limits of online learning algorithm first appeared in a series of work that mostly came from the statistical physics communities [3, 5, 13–16] in the 1990s. Similar to our setting, those early papers studied the dynamics of various online learning algorithms in high dimensions. In particular, they show that the mean-squared error (MSE) of the estimation, together with a few other "order parameters", can be characterized as the solution of a deterministic system of coupled ordinary differential equations (ODEs) in the large system limit. One limitation of such ODE-level analysis is that it cannot provide information about the distributions of the estimates. The latter are often needed when one wants to understand more general performance metrics beyond the MSE. Another limitation is that the ODE analysis cannot handle cases where the algorithms have non-quadratic regularization terms (*e.g.*, the incorporation of $\ell_1$ norms to promote sparsity.) In this paper, we show that both limitations can be eliminated by using our PDE-level analysis, which tracks the asymptotic evolution of the probability distributions of the estimates given by the algorithm. In a recent paper [10], the dynamics of an ICA algorithm was studied via a diffusion approximation. As an important distinction, the analysis in [10] keeps the ambient dimension $n$ *fixed* and studies the scaling limit of the algorithm as the step size tends to 0. The resulting PDEs involve $\mathcal{O}(n)$ spatial variables. In contrast, our analysis studies the limit as the dimension $n \to \infty$, with a constant step size. The resulting PDEs only involve 2 spatial variables. This low-dimensional characterization makes our limiting results more practical to use, especially when the dimension is large.

The basic idea underlying our analysis can trace its root to the early work of McKean [17, 18], who studied the statistical mechanics of Markovian-type mean-field interactive particles. The mathematical foundation of this line of research has been further established in the 1980s (see, e.g., [19, 20]). This theoretical tool has been used in the analysis of high-dimensional MCMC algorithms [21]. In our work, we study algorithms through the lens of high-dimensional stochastic processes. Interestingly, the analysis does not explicitly depend on whether the underlying optimization problem is convex or nonconvex. This feature makes the presented analysis techniques a potentially very useful tool in understanding the effectiveness of using low-complexity iterative algorithms for solving high-dimensional nonconvex estimation problems, a line of research that has recently attracted much attention (see, e.g., [22–25].)

The rest of the paper is organized as follows. We first describe in Section 2 the observation model and the online ICA algorithm studied in this work. The main convergence results are given in Section 3, where we show that the time-varying joint empirical measure of the target independent component and its estimates given by the algorithm can be characterized, in the high-dimensional limit, by the solution of a deterministic PDE. Due to space constraint, we only provide in the appendix a formal derivation leading to the PDE, and leave the rigorous proof of the convergence to a follow-up paper. Finally, in Section 4 we present some insight obtained from our asymptotic analysis. In particular, in the high-dimensional limit, the original coupled dynamics associated with the algorithm will be asymptotically "decoupled", with each coordinate independently solving a 1-D effective minimization problem via stochastic gradient descent.

**Notations and Conventions:**  Throughout this paper, we use boldfaced lowercase letters, such as $\boldsymbol{\xi}$ and $\boldsymbol{x}_k$, to represent $n$-dimensional vectors. The subscript $k$ in $\boldsymbol{x}_k$ denotes the discrete-time iteration step. The $i$th component of the vectors $\boldsymbol{\xi}$ and $\boldsymbol{x}_k$ are written as $\xi_i$ and $x_{k,i}$, respectively.

## 2  Data model and online ICA

We consider a generative model where a stream of sample vectors $\boldsymbol{y}_k \in \mathbb{R}^n$, $k = 1, 2, \dots$ are generated according to

$$\boldsymbol{y}_k = \tfrac{1}{\sqrt{n}}\boldsymbol{\xi}c_k + \boldsymbol{a}_k, \tag{1}$$

where $\boldsymbol{\xi} \in \mathbb{R}^n$ is a unique feature vector we want to recover. (For simplicity, we consider the case of recovering a single feature vector, but our analysis technique can be generalized to study cases involving a finite number of feature vectors.) Here $c_k \in \mathbb{R}$ is an i.i.d. random variable drawn from an unknown non-Gaussian distribution $P_c$ with zero mean and unit variance. And $\boldsymbol{a}_k \sim \mathcal{N}(0, \boldsymbol{I} - \tfrac{1}{n}\boldsymbol{\xi}\boldsymbol{\xi}^T)$

models background noise. We use the normalization $\|\boldsymbol{\xi}\|^2 = n$ so that in the large $n$ limit, all elements $\xi_i$ of the vector are $\mathcal{O}(1)$ quantities. The observation model (1) is equivalent to the standard sphered data model $\boldsymbol{y}_k = \boldsymbol{A} \begin{bmatrix} c_k \\ \boldsymbol{s}_k \end{bmatrix}$, where $\boldsymbol{A} \in \mathbb{R}^{n \times n}$ is an orthonormal matrix with the first column being $\boldsymbol{\xi}/\sqrt{n}$ and $\boldsymbol{s}_k$ is an i.i.d. $(n-1)$-dimensional standard Gaussian random vector.

To establish the large $n$ limit, we shall assume that the empirical measure of $\boldsymbol{\xi}$ defined by $\mu(\xi) = \frac{1}{n} \sum_{i=1}^{n} \delta(\xi - \xi_i)$ converges weakly to a deterministic measure $\mu^*(\xi)$ with finite moments. Note that this assumption can be satisfied in a stochastic setting, where each element of $\boldsymbol{\xi}$ is an i.i.d. random variable drawn from $\mu^*(\xi)$, or in a deterministic setting [*e.g.*, $\boldsymbol{\xi}_i = \sqrt{2}(i \bmod 2)$, in which case $\mu^*(\xi) = \frac{1}{2}\delta(\xi) + \frac{1}{2}\delta(\xi - \sqrt{2})$.]

We use an online learning algorithm to extract the non-Gaussian component $\boldsymbol{\xi}$ from the data stream $\{\boldsymbol{y}_k\}_{k \geq 1}$. Let $\boldsymbol{x}_k$ be the estimate of $\boldsymbol{\xi}$ at step $k$. Starting from an initial estimate $\boldsymbol{x}_0$, the algorithm update $\boldsymbol{x}_k$ by

$$\widetilde{\boldsymbol{x}}_k = \boldsymbol{x}_k + \tfrac{\tau_k}{\sqrt{n}} f(\tfrac{1}{\sqrt{n}} \boldsymbol{y}_k^T \boldsymbol{x}_k) \boldsymbol{y}_k - \tfrac{\tau_k}{n} \phi(\boldsymbol{x}_k)$$
$$\boldsymbol{x}_{k+1} = \tfrac{\sqrt{n}}{\|\widetilde{\boldsymbol{x}}_k\|} \widetilde{\boldsymbol{x}}_k, \tag{2}$$

where $f(\cdot)$ is a given twice differentiable function and $\phi(\cdot)$ is an element-wise nonlinear mapping introduced to enforce prior information about $\boldsymbol{\xi}$, *e.g.*, sparsity. The scaling factor $\frac{1}{\sqrt{n}}$ in the above equations makes sure that each component $\boldsymbol{x}_{k,i}$ of the estimate is of size $\mathcal{O}(1)$ in the large $n$ limit.

The above online learning scheme can be viewed as a projected stochastic gradient algorithm for solving an optimization problem

$$\min_{\|\boldsymbol{x}\|=n} -\frac{1}{K} \sum_{k=1}^{K} F(\tfrac{1}{\sqrt{n}} \boldsymbol{y}_k^T \boldsymbol{x}) + \frac{1}{n} \sum_{i=1}^{n} \Phi(x_i), \tag{3}$$

where $F(x) = \int f(x)\, \mathrm{d}x$ and

$$\Phi(x) = \int \phi(x)\, \mathrm{d}x \tag{4}$$

is a regularization function. In (2), we update $\boldsymbol{x}_k$ using an instantaneous noisy estimation $\frac{1}{\sqrt{n}} f(\frac{1}{\sqrt{n}} \boldsymbol{y}_k^T \boldsymbol{x}_k) \boldsymbol{y}_k$, in place of the true gradient $\frac{1}{K\sqrt{n}} \sum_{k=1}^{K} f(\frac{1}{\sqrt{n}} \boldsymbol{y}_k^T \boldsymbol{x}_k) \boldsymbol{y}_k$, once a new sample $\boldsymbol{y}_k$ is received.

In practice, one can use $f(x) = \pm x^3$ or $f(x) = \pm \tanh(x)$ to extract symmetric non-Gaussian signals (for which $\mathbb{E} c_k^3 = 0$ and $\mathbb{E} c_k^4 \neq 3$) and use $f(x) = \pm x^2$ to extract asymmetric non-Gaussian signals. The algorithm in (2) with $f(x) = x^3$ can also be regarded as implementing a low-rank tensor decomposition related to the empirical kurtosis tensor of $\boldsymbol{y}_k$ [10, 11].

For the nonlinear mapping $\phi(x)$, the choice of $\phi(x) = \beta x$ for some $\beta > 0$ corresponds to using an $L_2$ norm in the regularization term $\Phi(x)$. If the feature vector is known to be sparse, we can set $\phi(x) = \beta \operatorname{sgn}(x)$, which is equivalent to adding an $L_1$-regularization term.

## 3  Main convergence result

We provide an exact characterization of the dynamics of the online learning algorithm (2) when the ambient dimension $n$ goes to infinity. First, we define the joint empirical measure of the feature vector $\boldsymbol{\xi}$ and its estimate $\boldsymbol{x}_k$ as

$$\mu_t^n(\xi, x) = \frac{1}{n} \sum_{i=1}^{n} \delta(\xi - \xi_i, x - x_{k,i}) \tag{5}$$

with $t$ defined by $k = \lfloor tn \rfloor$. Here we rescale (*i.e.*, "accelerate") the time by a factor of $n$.

The joint empirical measure defined above carries a lot of information about the performance of the algorithm. For example, as both $\boldsymbol{\xi}$ and $\boldsymbol{x}_k$ have the same norm $\sqrt{n}$ by definition, the normalized correlation between $\boldsymbol{\xi}$ and $\boldsymbol{x}_k$ defined by

$$Q_t^n = \frac{1}{n} \boldsymbol{\xi}^T \boldsymbol{x}_k$$

can be computed as $Q_t^n = \mathbb{E}_{\mu_t^n}[\xi x]$, *i.e.*, the expectation of $\xi x$ taken with respect to the empirical measure. More generally, any separable performance metric $H_t^n = \frac{1}{n}\sum_{i=1}^n h(\xi_i, x_{k,i})$ with some function $h(\cdot, \cdot)$ can be expressed as an expectation with respect to the empirical measure $\mu_t^n$, *i.e.*, $H_t^n = \mathbb{E}_{\mu_t^n} h(\xi, x)$.

Directly computing $Q_t^n$ via the expectation $\mathbb{E}_{\mu_t^n}[\xi x]$ is challenging, as $\mu_t^n$ is a *random* probability measure. We bypass this difficulty by investigating the limiting behavior of the joint empirical measure $\mu_t^n$ defined in (5). Our main contribution is to show that, as $n \to \infty$, the sequence of random probability measures $\{\mu_t^n\}_n$ converges weakly to a deterministic measure $\mu_t$. Note that the limiting value of $Q_t^n$ can then be computed from the limiting measure $\mu_t^n$ via the identity $\lim_{n\to\infty} Q_t^n = \mathbb{E}_{\mu_t}[\xi x]$.

Let $P_t(x, \xi)$ be the density function of the limiting measure $\mu_t(\xi, x)$ at time $t$. We show that it is characterized as the unique solution of the following nonlinear PDE:

$$\frac{\partial}{\partial t}P_t(\xi, x) = -\frac{\partial}{\partial x}\left[\Gamma(x, \xi, Q_t, R_t)P_t(\xi, x)\right] + \frac{1}{2}\Lambda(Q_t)\frac{\partial^2}{\partial x^2}P_t(\xi, x) \tag{6}$$

with

$$Q_t = \iint_{\mathbb{R}^2} \xi x P_t(\xi, x)\,\mathrm{d}x\,\mathrm{d}\xi \tag{7}$$

$$R_t = \iint_{\mathbb{R}^2} x\phi(x) P_t(\xi, x)\,\mathrm{d}x\,\mathrm{d}\xi \tag{8}$$

where the two functions $\Lambda(Q)$ and $\Gamma(x, \xi, Q, R)$ are defined as

$$\Lambda(Q) = \tau^2 \left\langle f^2\left(cQ + e\sqrt{1-Q^2}\right)\right\rangle \tag{9}$$

$$\Gamma(x, \xi, Q, R) = x\left[QG(Q) + \tau R - \tfrac{1}{2}\Lambda(Q)\right] - \xi G(Q) - \tau\phi(x) \tag{10}$$

where

$$G(Q) = -\tau\left\langle f\left(cQ + e\sqrt{1-Q^2}\right)c\right\rangle + \tau Q\left\langle f'\left(cQ + e\sqrt{1-Q^2}\right)\right\rangle. \tag{11}$$

In the above equations, $e$ and $c$ denote two independent random variables, with $e \sim \mathcal{N}(0,1)$ and $c \sim P_c$, the non-Gaussian distribution of $c_k$ introduced in (2); the notation $\langle\cdot\rangle$ denotes the expectation over $e$ and $c$; and $f(\cdot)$ and $\phi(\cdot)$ are the two functions used in the online learning algorithm (2).

When $\phi(x) = 0$ (and therefore $R_t = 0$), we can derive a simple ODE for $Q_t$ from (6) and (7):

$$\frac{\mathrm{d}}{\mathrm{d}t}Q_t = (Q_t^2 - 1)G(Q_t) - \frac{1}{2}Q_t\Lambda(Q_t).$$

**Example 1** As a concrete example, we consider the case when $c_k$ is drawn from a symmetric non-Gaussian distribution. Due to symmetry, $\mathbb{E}\,c_k^3 = 0$. Write $\mathbb{E}\,c_k^4 = m_4$ and $\mathbb{E}\,c_k^6 = m_6$. We use $f(x) = x^3$ in (2) to detect the feature vector $\boldsymbol{\xi}$. Substituting this specific $f(x)$ into (9) and (11), we obtain

$$G(Q) = \tau Q^3(m_4 - 3) \tag{12}$$

$$\Lambda(Q) = \tau^2\left[15 + 15Q^4(1-Q^2)(m_4 - 3) + Q^6(m_6 - 15)\right] \tag{13}$$

and $\Gamma(x, \xi, Q, R)$ can be computed by substituting (12) and (13) into (10). Moreover, for the case $\phi(x) = 0$, we derive a simple ODE for $q_t = Q_t^2$ as

$$\frac{\mathrm{d}q_t}{\mathrm{d}t} = -2\tau_t q_t^2(1 - q_t)(m_4 - 3) - \tau_t^2 q_t\left[15q_t^2(1 - q_t)(m_4 - 3) + q_t^3(m_6 - 15) + 15\right]. \tag{14}$$

Numerical verifications of the ODE results are shown in Figure 1(a). In our experiment, the ambient dimension is set to $n = 5000$ and we plot the averaged results as well as error bars (corresponding to one standard deviation) over 10 independent trials. Two different initial values of $q_0 = Q_0^2$ are used. In both cases, the asymptotic theoretical predictions match the numerical results very well.

The ODE in (14) can be solved analytically. Next we briefly discuss its stability. The right-hand side of (14) is plotted in Figure 1(b) as a function of $q_t$. It is clear that the ODE (14) always admits a

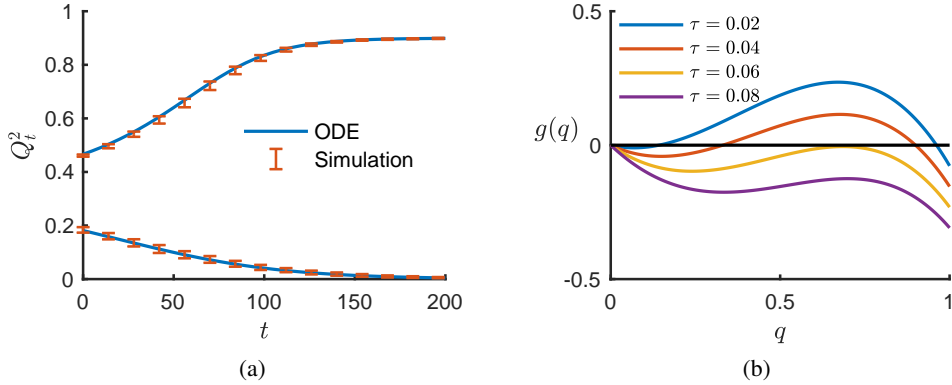

(a)                                                    (b)

Figure 1: (a) Comparison between the analytical prediction given by the ODE in (14) with numerical simulations of the online ICA algorithm. We consider two different initial values for the algorithm. The top one, which starts from a better initial guess, converges to an informative estimation, whereas the bottom one, with a worse initial guess, converges to a non-informative solution. (b) The stability of the ODE in (14). We draw $g(q) = \frac{1}{\tau}\frac{dq}{dt}$ for different value of $\tau = 0.02, 0.04, 0.06, 0.08$ from top to bottom.

solution $q_t = 0$, which corresponding to a trivial, non-informative solution. Moreover, this trivial solution is always a stable fixed point. When the stepsize $\tau > \tau_c$ for some constant $\tau_c$, $q_t = 0$ is also the unique stable fixed point. When $\tau < \tau_c$ however, two additional solutions of the ODE emerge. One is a stable fixed point denoted by $q_S^*$ and the other is an unstable fixed point denoted by $q_u^*$, with $q_u^* < q_S^*$. Thus, in order to reach an informative solution, one must initialize the algorithm with $Q_0^2 > q_u^*$. This insight agrees with a previous stability analysis done in [26], where the authors investigated the dynamics near $q_t = 0$ via a small $q_t$ expansion.

**Example 2** In this experiment, we verify the accuracy of the asymptotic predictions given by the PDE (6). The settings are similar to those in Example 1. In addition, we assume that the feature vector $\boldsymbol{\xi}$ is sparse, consisting of $\rho n$ nonzero elements, each of which is equal to $1/\sqrt{\rho}$. Figure 2 shows the asymptotic conditional density $P_t(x|\xi)$ for $\xi = 0$ and $\xi = 1/\sqrt{\rho}$ at two different times. These theoretical predictions are obtained by solving the PDE (6) numerically. Also shown in the figure are the empirical conditional densities associated with one realization of the ICA algorithm. Again, we observe that the theoretical predictions and numerical results have excellent agreement.

To demonstrate the usefulness of the PDE analysis in providing detailed information about the performance of the algorithm, we show in Figure 3 the performance of sparse support recovery using a simple hard-thresholding scheme on the estimates provided by the algorithm. By changing the threshold values, one can have trade-offs between the true positive and false positive rates. As we can see from the figure, this precise trade-off can be accurately predicted by our PDE analysis.

## 4 Insights given by the PDE analysis

In this section, we present some insights that can be gained from our high-dimensional analysis. To simplify the PDE in (6), we can assume that the two functions $Q_t$ and $R_t$ in (7) and (8) are given to us in an oracle way. Under this assumption, the PDE (6) describes the limiting empirical measure of the following stochastic process

$$z_{k+1,i} = z_{k,i} + \frac{1}{n}\Gamma(z_{k,i}, \xi_i, Q_{k/n}, R_{k/n}) + \sqrt{\frac{\Lambda(Q_{k/n})}{n}}w_{k,i}, \quad i = 1, 2, \ldots n \qquad (15)$$

where $w_{k,i}$ is a sequence of independent standard Gaussian random variables. Unlike the original online learning update equation (2) where different coordinates of $\boldsymbol{x}_k$ are coupled, the above process is *uncoupled*. Each component $z_{k,i}$ for $i = 1, 2, \ldots, n$ evolves independently when conditioned on $Q_t$ and $R_t$. The continuous-time limit of (15) is described by a stochastic differential equation (SDE)

$$dZ_t = \Gamma(Z_t, \xi, Q_t, R_t)\,dt + \sqrt{\Lambda(Q_t)}\,dB_t,$$

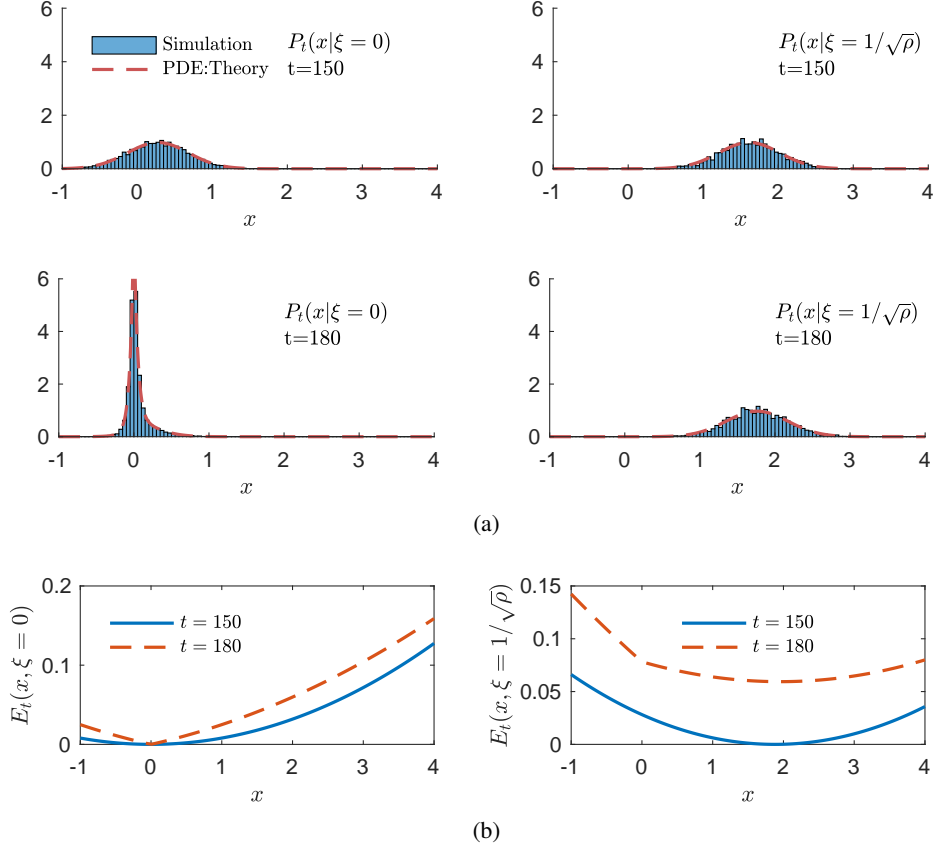

(a)

(b)

Figure 2:  (a) A demonstration of the accuracy of our PDE analysis. See the discussions in Example 2 for details. (b) Effective 1-D cost functions.

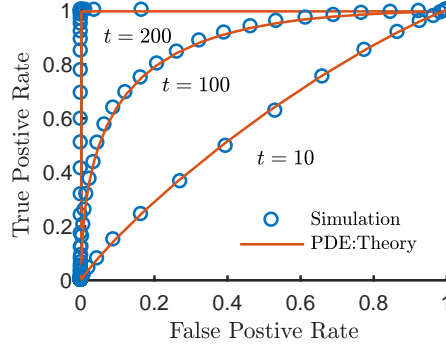

Figure 3:  Trade-offs between the true positive and false positive rates in sparse support recovery. In our experiment, $n = 10^4$, and the sparsity level is set to $\rho = 0.3$. The theoretical results obtained by our PDE analysis can accurately predict the actual performance at any run-time of the algorithm.

where $B_t$ is the standard Brownian motion.

We next have a closer look at the equation (15). Given a scalar $\xi$, $Q_t$ and $R_t$, we can define a time-varying 1-D regularized quadratic optimization problem $\min_{x \in \mathbb{R}} E_t(x, \xi)$ with the effective potential

$$E_t(x, \xi) = \tfrac{1}{2} d_t (x - b_t \xi)^2 + \tau \Phi(x), \tag{16}$$

where $d_t = Q_t G(Q_t) - \tfrac{1}{2} \Lambda(Q_t) + \tau R_t$ , $b_t = G(Q_t)/d_t$ and $\Phi(x)$ is the regularization term defined in (4). Then, the stochastic process (15) can be viewed as a stochastic gradient descent

for solving this 1-D problem with a step-size equal to $1/n$. One can verify that the exact gradient of (16) is $-\Gamma(x, \xi, Q_t, R_t)$. The third term $\sqrt{\frac{\Lambda(Q_k)}{n}} w_k$ in (15) adds stochastic noise to the true gradient. Interestingly, although the original optimization problem (3) is non-convex, its 1-D effective optimization problem is always convex for convex regularizers $\Phi(x)$ (*e.g.*, $\Phi(x) = \beta |x|$.) This provides an intuitive explanation for the practical success of online ICA.

To visualize this 1-D effective optimization problem, we plot in Figure 2(b) the effective potential $E_t(x, \xi)$ at $t = 0$ and $t = 100$, respectively. From Figure (2), we can see that the $L_1$ norm always introduces a bias in the estimation for all non-zero $\xi_i$, as the minimum point in the effective 1-D cost function is always shifted towards the origin. It is hopeful that the insights gained from the 1-D effective optimization problem can guide the design of a better regularization function $\Phi(x)$ to achieve smaller estimation errors without sacrificing the convergence speed. This may prove an interesting line of future work.

This uncoupling phenomenon is a typical consequence of mean-field dynamics, *e.g.*, the Sherrington-Kirkpatrick model [27] in statistical physics. Similar phenomena are observed or proved in other high dimensional algorithms especially those related to approximate message passing (AMP) [28–30]. However, for these algorithms using batch updating rules with the Onsager reaction term, the limiting densities of iterands are Gaussian. Thus the evolution of such densities can be characterized by tracking a few scalar parameters in discrete time. For our case, the limiting densities are typically non-Gaussian and they cannot be parametrized by finitely many scalars. Thus the PDE limit (6) is required.

## Appendix: A Formal derivation of the PDE

In this appendix, we present a *formal* derivation of the PDE (6). We first note that $(\boldsymbol{x}_k, \boldsymbol{\xi}_k)_k$ with $\boldsymbol{\xi}_k = \boldsymbol{\xi}$ forms an *exchangeable* Markov chain on $\mathbb{R}^{2n}$ driven by the random variable $c_k \sim P_c$ and the Gaussian random vector $\boldsymbol{a}_k$. The drift coefficient $\Gamma(x, \xi, Q, R)$ and the diffusion coefficient $\Lambda(Q)$ in the PDE (6) are determined, respectively, by the conditional mean and variance of the increment $x_{k+1,i} - x_{k,i}$, conditioned upon the previous state vector $\boldsymbol{x}_k$ and $\boldsymbol{\xi}_k$.

Let the increment of the gradient-descent step in the learning rule (2) be

$$\widetilde{\Delta}_{k,i} = \widetilde{x}_{k,i} - x_{k,i} = \tfrac{\tau_k}{\sqrt{n}} f(\tfrac{1}{\sqrt{n}} \boldsymbol{y}_k^T \boldsymbol{x}_k) y_{k,i} - \tfrac{\tau_k}{n} \phi(x_{k,i}) \tag{17}$$

where $\widetilde{x}_{k,i}$ is the $i$th component of the output $\widetilde{\boldsymbol{x}}_k$. Let $\mathbb{E}_k$ denote the conditional expectation with respect to $c_k$ and $\boldsymbol{a}_k$ given $\boldsymbol{x}_k$ and $\boldsymbol{\xi}_k$.

We first compute $\mathbb{E}_k\left[\widetilde{\Delta}_{k,i}\right]$ and $\mathbb{E}_k\left[\widetilde{\Delta}_{k,i}^2\right]$. From (1) and (17) we have

$$\mathbb{E}_k\left[\widetilde{\Delta}_{k,i}\right] = \tfrac{\tau_k}{\sqrt{n}} \mathbb{E}_k\left[f(Q_k^n c_k + \widetilde{e}_{k,i} + \tfrac{1}{\sqrt{n}} a_{k,i} x_{k,i})(\tfrac{1}{\sqrt{n}} \xi_i c_k + a_{k,i})\right] - \tfrac{\tau_k}{n} \phi(x_{k,i}),$$

where $Q_k^n = \tfrac{1}{n} \boldsymbol{\xi}^T \boldsymbol{x}_k$ and $\widetilde{e}_{k,i} = \tfrac{1}{\sqrt{n}}\left(\boldsymbol{a}_k^T \boldsymbol{x}_k - a_{k,i} x_{k,i}\right)$. We use the Taylor expansion of $f$ around $Q_k^n c_k + \widetilde{e}_{k,i}$ up to the first order and get

$$\mathbb{E}_k\left[f(Q_k^n c_k + \widetilde{e}_{k,i} + \tfrac{1}{\sqrt{n}} a_{k,i} x_{k,i})(\tfrac{1}{\sqrt{n}} \xi_i c_k + a_{k,i})\right]$$

$$= \mathbb{E}_k\left[f(Q_k^n c_k + \widetilde{e}_{k,i})(\tfrac{1}{\sqrt{n}} \xi_i c_k + a_{k,i})\right] + \tfrac{1}{\sqrt{n}} x_{k,i} \mathbb{E}_k\left[f'(Q_k^n c_k + \widetilde{e}_{k,i})(\tfrac{1}{\sqrt{n}} \xi_i c_k + a_{k,i}) a_{k,i}\right] + \delta_{k,i},$$

where $\delta_{k,i}$ includes all higher order terms. As $n \to \infty$, the random variable $Q_k^n$ converges to a deterministic quantity $Q_k$. Moreover, $\widetilde{e}_{k,i}$ and $a_{k,i}$ are both zero-mean Gaussian with the covariance matrix $\begin{bmatrix} 1 - Q_k^2 + O(\frac{1}{n}) & -\frac{1}{\sqrt{n}} \xi_{k,i} Q_k \\ -\frac{1}{\sqrt{n}} \xi_{k,i} Q_k & 1 + O(\frac{1}{n}) \end{bmatrix}$. We thus have

$$\mathbb{E}_k\left[f'(Q_k^n c_k + \widetilde{e}_{k,i})(\tfrac{1}{\sqrt{n}} \xi_i c_k + a_{k,i}) a_{k,i}\right] = \left\langle f'(Q_k c + \sqrt{1 - Q_k^2} e) \right\rangle + o(1)$$

and

$$\mathbb{E}_k \left[ f(Q_k^n c_k + \widetilde{e}_{k,i})(\tfrac{1}{\sqrt{n}} \xi_i c_k + a_{k,i}) \right]$$

$$= \left\langle f(Q_k c + \sqrt{1 - Q_k^2} e - \tfrac{\xi_i}{\sqrt{n}} Q_k a)(\tfrac{1}{\sqrt{n}} \xi_i c + a) \right\rangle$$

$$= \tfrac{1}{\sqrt{n}} \xi_i \left[ \left\langle cf(Q_k c + \sqrt{1 - Q_k^2} e) \right\rangle - Q_k \left\langle f'(Q_k c + \sqrt{1 - Q_k^2} e) \right\rangle \right] + o(\tfrac{1}{\sqrt{n}}),$$

where in the last line, we use the Taylor expansion again to expand $f$ around $Q_k c + \sqrt{1 - Q_k^2} e$ and the bracket $\langle \cdot \rangle$ denotes the average over two independent random variables $c \sim P_c$ and $e \sim \mathcal{N}(0,1)$. Thus, we have

$$\mathbb{E}_k \left[ \widetilde{\Delta}_{k,i} \right] = \frac{1}{n} \left[ -\xi_i G(Q_k) + \tau_k x_{k,i} \left\langle f'(Q_k c + \sqrt{1 - Q_k^2} e) \right\rangle - \tau_k \phi(x_{k,i}) \right] + o(\tfrac{1}{n}),$$

where the function $G(Q)$ is defined in (11).

To compute the (conditional) variance, we have

$$\mathbb{E}_k \left[ \widetilde{\Delta}_{k,i}^2 \right] = \tfrac{\tau_k^2}{n} \mathbb{E}_k \left[ f^2(Q_k^n + \widetilde{e}_{k,i}) \right] + o(\tfrac{1}{n}) = \tfrac{\tau_k^2}{n} \left\langle f^2(Q_k c + \sqrt{1 - Q_k^2} e) \right\rangle + o(\tfrac{1}{n}).$$

Next, we deal with the normalization step. Again, we use the Taylor expansion for the term $\left\| \tfrac{1}{n} \widetilde{\boldsymbol{x}}_k \right\|^{-1} = \left\| \tfrac{1}{n} \left( \boldsymbol{x}_k + \widetilde{\boldsymbol{\Delta}}_k \right) \right\|^{-1}$ up to the first order, which yields

$$\boldsymbol{x}_{k+1} = \boldsymbol{x}_k - \tfrac{1}{n} \boldsymbol{x}_k \left( \boldsymbol{x}_k^T \widetilde{\boldsymbol{\Delta}}_k + \tfrac{1}{2} \widetilde{\boldsymbol{\Delta}}_k^T \widetilde{\boldsymbol{\Delta}}_k \right) + \widetilde{\boldsymbol{\Delta}}_k + \boldsymbol{\delta}_k,$$

where $\boldsymbol{\delta}_k$ includes all higher order terms. Note that $\tfrac{1}{n} \boldsymbol{x}_k^T \widetilde{\boldsymbol{\Delta}}_k \approx \tfrac{1}{n} \sum_{i=1}^n x_{k,i} \mathbb{E}_k \left[ \widetilde{\Delta}_{k,i} \right]$, $\tfrac{1}{n} \widetilde{\boldsymbol{\Delta}}_k^T \widetilde{\boldsymbol{\Delta}}_k \approx \tfrac{1}{n} \sum_{i=1}^n \mathbb{E}_k \left[ \widetilde{\Delta}_{k,i}^2 \right]$ and $\tfrac{1}{n} \boldsymbol{x}_k^T \phi(\boldsymbol{x}) = R_k^n \to R_k$, we have

$$\mathbb{E}_k \left[ x_{k+1,i} - x_{k,i} \right] = \tfrac{1}{n} \Gamma(x_{k,i}, \xi_i, Q_k, R_k) + o(\tfrac{1}{n}).$$

Finally, the normalization step does not change the variance term, and thus

$$\mathbb{E}_k \left[ \left( x_{k+1,i} - x_{k,i} \right)^2 \right] = \mathbb{E}_k \left[ \widetilde{\Delta}_{k,i}^2 \right] + o(\tfrac{1}{n}) = \tfrac{1}{n} \Lambda(Q_k) + o(\tfrac{1}{n}).$$

The above computation of $\mathbb{E}_k(x_{k+1,i} - x_{k,i})$ and $\mathbb{E}_k(x_{k+1,i} - x_{k,i})^2$ connects the dynamics (2) to (15). In fact, both (2) and (15) have the same limiting empirical measure described by (6).

A rigorous proof of our asymptotic result is built on the weak convergence approach for measure-valued processes. Details will be presented in an upcoming paper. Here we only provide a sketch of the general proof strategy: First, we prove the tightness of the measure-valued stochastic process $(\mu_t^n)_{0 \le t \le T}$ on $D([0,T], \mathcal{M}(\mathbb{R}^2))$, where $D$ denotes the space of càdlàg processes taking values from the space of probability measures. This then implies that any sequence of the measure-valued process $\{(\mu_t^n)_{0 \le t \le T}\}_n$ (indexed by $n$) must have a weakly converging subsequence. Second, we prove any converging (sub)sequence must converge weakly to a solution of the weak form of the PDE (6). Third, we prove the uniqueness of the solution of the weak form of the PDE (6) by constructing a contraction mapping. Combining these three statements, we can then conclude that any sequence must converge to this unique solution.

**Acknowledgments** This work is supported by US Army Research Office under contract W911NF-16-1- 0265 and by the US National Science Foundation under grants CCF-1319140 and CCF-1718698.

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
