[Reviews · NeurIPS 2017]

Reviewer 1



The paper studies the high-dimensional scaling limit of a stochastic update algorithm for online Independent Component Analysis. The main result of the paper is an exact characterization of the evolution of the joint empirical distribution of the estimates output by the algorithm and the signal to be recovered, when the number of observations scales linearly with the dimension of the problem. The authors argue that in the limit, this joint distribution is the unique solution to a certain partial differential equation, from which the performance of the algorithm can be predicted, in accordance with the provided simulations. 1- Overall, the result is fairly novel, and provides interesting insight into the behavior of stochastic algorithms in this non-convex problem. The paper is written in a fairly clear manner and is (mostly) easy to read. My main concern is that the mathematics are written in a very informal way, so that it is not clear what the authors have actually proved. There are no theorem or proposition statements. E.g., under what conditions does the empirical measure have a weak limit? Under what (additional) conditions does this limit have a density? And when is this density the unique solution to the PDE? The heuristic justification in the appendix is not very convincing either (see bullet 4). 2- A very interesting insight of the theory is that the p variables of the problem decouple in the limit, where each variable obeys a diffusion equation independently of the others. The only global aspect retained by these local dynamics is via the order parameters R and Q. This phenomenon is standard in mean-field models of interacting particle systems, where the dynamics of the particles decouple when the order parameters converge to a limit. The authors should probably draw this connection explicitly; the relevant examples are the Sherrington-Kirkpatrick model, rank-one estimation in spiked models, compressed sensing, high-dim. robust estimation… 3- The connection to the SDE line 175 should be immediate given that the PDE (5) is its associated Fokker-Planck equation. The iteration (14) is easier seen as a discretization of the SDE rather than as some approximate dynamics solving the PDE. Therefore, the SDE should come immediately after the statement of the PDE, then the iteration. 4- The derivations in the appendix proceed essentially by means of a cavity (or a leave-one-out) method, but end abruptly without clear conclusion. The conclusion seems to be the decoupled iteration (14), not the actual PDE. This should be made clear, or otherwise, the derivation should be conducted further to the desired conclusion. The derivation contain some errors and typos; I couldn't follow the computations (e.g., not clear how line 207 was obtained, a factor c_k should appear next to Q_k^p on many lines...) 5-There are several imprecisions/typos in the notation: line 36: support recover*y* eq 2: sign inconsistency in front of \tau_k (- gradient, not +) line 100: x_k should be x line 133: u should be \xi (in many later occurrences) eq 8: \tau not defined (is it the limit of the step sizes \tau_k?) line 116: t = [k/p] should be k = [tp]

Reviewer 2



The article gives a scaling limit analysis of the ICA algorithm. The scaling limit is obtained in the regime where the dimension "p" of the vector to be recovered goes to infinity -- the scaling limit is a PDE that describes the evolution of the probability density of the joint distribution between "x" and "xi". The sketch of proof is convincing and standard (computation of mean/variance <--> drift/volatility), although I have not checked all the details. The most interesting part is indeed the insights one can gain through this scaling analysis. 1) in some particular cases, the analysis shows that it is necessary to have small enough learning rate and a "good" initial guess in order to converge to a non-trivial solution. That's extremely interesting and gives theoretical justification for empirical observations previously obtained in the literature. 2) in other particular cases, the analysis explains why (by introducing a convex "effective" objective function) online ICA is empirically successful even though minimizing a non-convex objective function.

Reviewer 3



\section*{Summary} The paper looks at an online learning method for a generative model as give in equation (1) in the paper, with the unknown being the $p$-dimensional parameter $\xi$. The online learning algorithm is described in equation (2). The main contribution of the paper is that it provides a PDE limit for an empirical measure (in particular for its transition density) involving the co-ordinates of $\xi$ and the iterative estimate $x_{k}$ - after appropriate time rescaling. The authors argue that getting such an explicit limit allows for a better understanding of the performance of the algorithm in high dimensions. \section*{Comments} I am afraid the content of the paper is beyond my areas of expertise. I do appreciate that similar results have proven to be useful in the context of high-dimensional MCMC algorithms, so I would guess that the direction followed in the paper is of interest.